# In Vivo Analysis of a Biodegradable Magnesium Alloy Implant in an Animal Model Using Near-Infrared Spectroscopy

**DOI:** 10.3390/s23063063

**Published:** 2023-03-13

**Authors:** Anna Mathew, Hafiz Wajahat Hassan, Olga Korostynska, Frank Westad, Eduarda Mota-Silva, Luca Menichetti, Peyman Mirtaheri

**Affiliations:** 1Faculty of Technology, Art and Design, Department of Mechanical, Electronic and Chemical Engineering, OsloMet—Oslo Metropolitan University, 0130 Oslo, Norway; 2Department of Engineering Cybernetics, Norwegian University of Science and Technology, 7034 Trondheim, Norway; 3Institute of Clinical Physiology, National Research Council (IFC-CNR), San Cataldo Research Area, 56124 Pisa, Italy; 4Institute of Life Sciences, Sant’Anna School of Advanced Studies, 56127 Pisa, Italy

**Keywords:** biodegradable implants, magnesium implant, near-infrared spectroscopy, optical probe, principal component analysis, PCA, in situ pH

## Abstract

Biodegradable magnesium-based implants offer mechanical properties similar to natural bone, making them advantageous over nonbiodegradable metallic implants. However, monitoring the interaction between magnesium and tissue over time without interference is difficult. A noninvasive method, optical near-infrared spectroscopy, can be used to monitor tissue’s functional and structural properties. In this paper, we collected optical data from an in vitro cell culture medium and in vivo studies using a specialized optical probe. Spectroscopic data were acquired over two weeks to study the combined effect of biodegradable Mg-based implant disks on the cell culture medium in vivo. Principal component analysis (PCA) was used for data analysis. In the in vivo study, we evaluated the feasibility of using the near-infrared (NIR) spectra to understand physiological events in response to magnesium alloy implantation at specific time points (Day 0, 3, 7, and 14) after surgery. Our results show that the optical probe can detect variations in vivo from biological tissues of rats with biodegradable magnesium alloy “WE43” implants, and the analysis identified a trend in the optical data over two weeks. The primary challenge of in vivo data analysis is the complexity of the implant interaction near the interface with the biological medium.

## 1. Introduction

Osteoporosis has been a concern in European countries for decades. Osteoporosis patients have a higher tendency for bone fractures. Worldwide, it is reported to have more than nine million fractures a year. It means somebody gets a fracture every three seconds [1]. Bone healing may take a few weeks and sometimes several months, impacting the quality of life. Infection post-surgery is challenging to handle, making patient life even harder [2]. Bone tissues are unique as they can heal themselves after damage. Depending on the nature of the fracture, at times, supporting frames such as plates and screws are inserted into the bone to avoid deformed healing. If supporting plates/screws are made of biodegradable materials, it would prevent further complications with additional surgical procedures to remove the implants in the same region [3,4]. When a biodegradable implant decays, the mechanical properties of the materials reduce with time through biochemical reactions. Magnesium inside the living body is balanced based on the dynamic equilibrium of absorption and excretion [5]. Elements within certain limits are permissible for the human body, favoring the use of such biodegradable magnesium implants [6], which also have good biocompatibility. Magnesium implants have properties such as an elastic modulus and density close to natural bone [3,7]. In addition, magnesium-based alloys can promote tissue regeneration [5]. About 60% of the total magnesium is stored in the bones [8]. Thus, magnesium’s benefits make it one of the most promising biodegradable substitutes for biomedical applications with numerous clinical trials [9].

Pure magnesium degrades quickly and does not provide adequate mechanical strength until bone healing occurs. Therefore, magnesium is typically alloyed to produce magnesium-based implants that are suitable for use in implant applications. It not only helps to prevent the fast degradation of magnesium, but also extends the mechanical life of the implant [3,10]. Magnesium alloy as an implant material has been researched to improve its properties [3,7,9,11,12]. Alloying helps to reduce the fast degradation rate of pure magnesium implants, giving sufficient mechanical strength to the bone until it heals completely [5,13]. To sum up, biodegradable implant reaction rates are to be such that they provide sufficient time for bone healing to serve the purpose.

There are chemical changes near the implant interface, as illustrated by the equations [7,9,12]. When magnesium reacts with water in the tissue, it produces hydrogen, as in Equation (4). Gas bubbles occupy loose skin near the magnesium implants [14]. The gas gradually disappears within 2–3 weeks of the in vivo implant surgery [9].
(1)$$\mathrm{Mg}\to {\mathrm{Mg}}^{2+}+2e^{-}\;\mathrm{The anodic reaction}$$
(2)$$2H_{2}O+2e^{-}\to \;2{\mathrm{OH}}^{-}+{\;H}_{2\;}↑\;\mathrm{The cathodic reaction}$$
(3)$${\mathrm{Mg}}^{2+}+2{\mathrm{OH}}^{-}\to \mathrm{Mg}{\left(\mathrm{OH}\right)}_{2}\;↓$$

Then the overall reaction can be written:(4)$$\mathrm{Mg}+2H_{2}O\to \mathrm{Mg}{\left(\mathrm{OH}\right)}_{2}\;↓+{\;H}_{2\;}↑\;\mathrm{Overall reaction}$$

These $\mathrm{Mg}{\left(\mathrm{OH}\right)}_{2}$ hydroxide layers (based on Equation (4)) form a layer covering the implant surface. However, its instability makes it react with the chloride ions from body fluids to form highly soluble magnesium chloride, as in Equation (5) [7]. The hydroxide ions (${\mathrm{OH}}^{-})$ that are formed near the interface of the implant influence the local pH.
(5)$$\mathrm{Mg}{\left(\mathrm{OH}\right)}_{2}+2{\mathrm{Cl}}^{-}\to {\mathrm{MgCl}}_{2}+2{\mathrm{OH}}^{-}$$

The pH of a medium is the concentration of hydrogen ions (H^+^). The exact pH can also be the ${\mathrm{OH}}^{-}$ ion concentration value. A solution is alkaline or basic if the increase in hydroxide ions ${\mathrm{OH}}^{-}$ is numerically the same as a decrease in (H^+^). The reaction presented in Equation (5) suggests that the local pH near the interface shall increase due to the expulsion of hydroxide ions ${\mathrm{OH}}^{-}$ [12,15]. The biochemical reactions are highly complex and form precipitates [16].
(6)$${\mathrm{OH}}^{-}+{\mathrm{HCO}}_{3}^{-}\to {\;\mathrm{CO}}_{3}^{2-}+{\;H}_{2\;}O$$
(7)$${\mathrm{Mg}}^{2+}+{\mathrm{CO}}_{3}^{2-}\to {\mathrm{MgCO}}_{3}\;↓$$
(8)$${\mathrm{HPO}}_{4}^{2-}+{\;\mathrm{OH}}^{-}\to {\;\mathrm{PO}}_{4}^{3-}+{\;H}_{2\;}O$$
(9)$$3{\mathrm{Mg}}^{2+}+2{\mathrm{PO}}_{4}^{3-}\;\to {\mathrm{Mg}}_{3}{\left({\mathrm{PO}}_{4}\right)}_{2\;}\;↓$$

Calcium also reacts to form precipitates:(10)$${\mathrm{Ca}}^{2+}+{\mathrm{CO}}_{3}^{2-}\to {\mathrm{CaCO}}_{3}\;↓$$
(11)$$3{\mathrm{Ca}}^{2+}+2{\mathrm{PO}}_{4}^{3-}\;\to {\mathrm{Ca}}_{3}{\left({\mathrm{PO}}_{4}\right)}_{2\;}\;↓$$

This chemical analysis indicates a delicate pH balance near the implant surroundings. A medium is alkaline or basic if there is an increase in hydroxide ions. As in Equation (5), the local pH near the interface shall increase due to the expulsion of hydroxide ions. Nevertheless, as the reactions proceed (Equations (7) and (9)–(11)), these hydroxide ions react and form precipitates, such as ${\mathrm{MgCO}}_{3},{\;\mathrm{Mg}}_{3}{\left({\mathrm{PO}}_{4}\right)}_{2},{\mathrm{CaCO}}_{3},{\mathrm{and}\;\mathrm{Ca}}_{3}{\left({\mathrm{PO}}_{4}\right)}_{2}$. Such precipitates near the interface can form a further coating near the implant that helps reduce the degradation [7,12]. The literature mentions the formation of phosphorous precipitates after a few weeks of implantation of magnesium implants in an in vivo analysis [5]. The precipitate layer obstructs the interaction of the implant with water and helps control the formation of hydroxide ions that contribute to pH changes. The body’s ability to naturally balance body parameters helps to suppress pH changes and maintains pH homeostasis. When the pH shifts toward alkalosis due to the magnesium implant’s presence, it can affect the healing of the wound caused by surgery [15]. It is pH homeostasis that helps to balance the change.

The reaction (from Equation (4)) happens with the support of water. In addition, the literature highlights that water is a byproduct of the succeeding reactions [7]. Thus, near the implant interface, there are changes related to the presence of water. In an in vivo system, water, blood, and other physiological fluids are naturally present. Thus, a magnesium implant in constant contact with fluids and tissues leads to a dynamic interface that constantly undergoes functional (metabolic) and structural (surface) changes in the implant–tissue interface.

Currently, monitoring implants requires a complex and resource-demanding healthcare system [17]. Near-infrared (NIR)/IR spectroscopy provides a patient-friendly, nondestructive, and noninvasive technology. As part of a previous project (MgSafe), an optical probe (Figure 1) was developed to collect photons within NIR/IR spectroscopic data from tissue samples [18,19,20]. The construction and validation of the probe are reported in [21,22], respectively. In this paper, we utilize optical spectral analysis to investigate the effects of magnesium implants in vitro and in vivo. The in vitro experiment aims to examine the optical interaction of a magnesium implant in a cell culture medium over time. In the in vivo setup, we perform exploratory data analysis to identify a specific observation or trend for animals with magnesium implants. Moreover, we discuss possible reasons for the differences in trends between the in vivo and in vitro experiments. We hypothesize that we can detect a difference in the optical spectra in animals implanted with a biodegradable Mg implant over time.

## 2. Materials and Methods

The optical probe emits light from 680 nm to 1100 nm. It can pass through tissues for measuring changes in biological tissues noninvasively [18,23]. The light from the source reaches the detector after passing through the medium of interest. Some photons in the light ray are absorbed in the medium while others are scattered. Hence, the light that reaches the detector indirectly indicates the changes in the medium through which it passes [24]. Experiments were performed as part of this publication, including in vivo and in vitro experiments. In vivo experiments were conducted on animals. Preparatory in vitro data collection and analysis were conducted to minimize uncertainties. The primary target of the experiments was to gain insights into the use of optical near-infrared spectroscopy to monitor the interaction of biodegradable Mg-based implants with tissue over time, enabling the possibility of grouping optical spectra on a particular day in the corresponding experiment. In vivo and in vitro experiments were studied in the first two weeks; however, two experiments were conducted in two different environments and were not compared directly day to day. In vitro experiments were designed to study the effects of magnesium degradation in simulated body fluid. In vivo experiments were conducted to evaluate the performance of magnesium pins implanted in rats. The time points were selected based on the goals and requirements of each experiment. The animals’ well-being was the priority; hence, day points varied slightly in this pilot study. The animals’ temperature was monitored during the experiments and ensured that it was stable and did not differ significantly for the animals. This paper investigates the observations or trends identified in the in vitro experiment in the context of the in vivo experiment.

### 2.1. Methodology

For the reflection-type measurement, the light source and detector were on the same side of the thick medium, such as hard tissues. Figure 2a illustrates the experimental setup used in vivo. The light source and detector were on opposite sides of the medium for the transmission-type measurement, as shown in Figure 2b. The spectrometer (Avaspec-2048x14, Avantes, Apeldoorn, The Netherlands), used with Avantes software AvaSoft8, produced a spectrum ranging from 650 nm to 1100 nm. The instrument was calibrated every time the experiment was set up for the measurements. Based on the degradation of the Mg implant in the surrounding medium, the reflection or transmission of light was affected, resulting in changes in the scattering and absorption of the light [21].

One of the efficient exploratory model approaches is principal component analysis (PCA). Dimensionality reduction based on the principal components is PCA’s main advantage as it helps to plot the data into their principal components. In spectral analysis, the variables are wavelengths, and the multi-dimensional nature of the spectral data makes it challenging to visualize. Therefore, PCA was performed using python with the help of the scikit-learn library on the collected data after preprocessing. The explained variance and the score plot were further analyzed and interpreted based on the different experimental conditions [26,27]. Furthermore, a comparison study was conducted based on the in vivo progressions results of two weeks post-surgery.

### 2.2. In Vitro Experiment and the Trend

The primary goal of the in vitro study was to check the feasibility of using variations in the optical spectrum as a response to changes in the surrounding medium caused by the degradation of magnesium implants with time. The experiment used the transmission-type optical data collection [25], as illustrated in Figure 2b. The experiment used three samples of biodegradable Mg alloy ZX00 disks [0.45 wt% Zn—0.45 wt% Ca. Tech (Bri.Tech, Graz, Austria)] in cell culture medium DMEM (Gibco Dulbecco’s Modified Eagle Medium). Optical data were collected on days 0, 2, 5, and 10. Day 0 refers to the spectrum collected before placing the ZX00 disk in the medium. Day 2 is the second day’s optical spectrum from the DMEM solution. Likewise, day 5 and day 10 give optical spectra for the corresponding days. Throughout the experiments, we observed a gradual change in the color of the DMEM solution as the pH levels varied with the degradation of the magnesium implants, although the change was not significant. We used DMEM with a composition of 1 g/L of glucose and sodium bicarbonate without L-glutamine for our experiments. Each day, ten measurements were performed on each Mg alloy disk. Data cleaning that involved removing noise, artifacts, and outliers that could affect the analysis results was followed by dimension reduction based on its principal components.

### 2.3. In Vivo Experiment

#### 2.3.1. Ethical Considerations

All in vivo experiments were carried out following the National Ethical Guidelines (Italian Ministry of Health, Rome, Italy; D.L.vo 26/2014) and the guidelines from Directive 2010/63/EU of the European Parliament. The protocol was approved by the Instituto Superiore di Sanità on behalf of the Italian Ministry of Health and Ethical Panel (Prot. no. 299/2020-PR) and the local ethics committee. Additionally, the protocol conformed to the ARRIVE guidelines.

#### 2.3.2. In Vivo Data Gathering

This study included four 12-week-old female Wistar rats that were implanted with Mg alloy WE43 cylindrical pins inserted through the mid-diaphyseal region of both femurs. Briefly, the 12-week-old female Wistar rats were anesthetized, and a transcortical hole was drilled in the femur using a 1.55 mm diameter drill. A pin implant made of Mg alloy (WE43) was inserted, and the wound site was closed with resorbable sutures. Please refer to our previous publication [28] for more information on the surgical procedure. NIRS acquisitions were performed at predetermined time points. In vivo experiments were conducted to evaluate the performance of magnesium pins implanted in rats and based on physiological meaning. The animals’ well-being was the priority; hence, day points were also selected considering the postoperative recovery state of the animals. For example, on day 3, animals fully recovered from the surgical procedure and had normal hydration levels. Day 0 acquisition was made immediately after surgery, so animals were already anesthetized. We used a rectal thermometer in all animals to guarantee their temperature stabilized at a normal value. An optical reflective setup was used to measure the NIR data (referring to Figure 2a) for two weeks after surgery. The probe was placed in a consistent manner for each measurement using the femur as an anatomical reference on the lateral side of the leg over the implantation site and parallel with the femur. Acquisitions were made on day 0 (a few minutes after surgery), day 3, day 7, and day 14 after the animals were anesthetized, and the anesthesia was maintained throughout the measurements. The animals were anesthetized with 2.5% isoflurane in pure oxygen and placed laterally on a stereo-foam platform to minimize heat loss. The animals woke up naturally 1–2 min after removing the anesthesia. We paid special attention to maintaining a controlled breathing rate. If necessary, any regrowth fur was removed with a depilatory cream (Veet) 2–3 min before acquisition to avoid any skin microcirculation changes due to friction. All acquisitions were made in the same room at a constant temperature of 25 °C and under low luminosity conditions to reduce background noise. More details on the acquisition procedure can be found in our previous publication [21].

The selected time points were chosen considering the critical pH variations in the surrounding region of the implant during the first two weeks [25] and physiological meaning. The first two weeks are determinant for the success of implant osseointegration, and it is when soft tissue surrounding the implant heals. Between day 0 and days 2–3, there is an ongoing process of acute inflammation, blood vessels ruptured during the surgery, leading to hematoma formation. Several complex biochemical interactions happen around the implantation site. At the same time, when the implant degradation rate is fastest, the fast release of hydrogen leads to gas bubble formation. From days 5 to 7, it is possible to find new tissue organization, i.e., revascularization, new blood vessels forming, bone tissue regenerating, and gas bubbles reducing. Other in vivo studies investigating biodegradable implant degradation and surrounding tissue regeneration use similar time points. Hence, the selected time points do not differ significantly as the degradation rate of the alloys in vitro is different from in vivo [29].

Throughout this study, the animals’ physiological parameters, such as temperature and blood oxygen saturation levels, were closely monitored for any signs of distress or discomfort, including during surgery and NIRS acquisitions. Although we did not use instruments to monitor breathing rates in real time, we manually checked the animals’ breathing rates to ensure their overall health. Since our focus was mainly on deep-tissue measurements, we anticipated that respiration’s impact on NIRS measurements would be minimal. Our findings are supported by the analysis of oxy and de-oxy data, which revealed that changes in respiration rate would result in rapid changes in de-oxy measurements instead of oxy values. Additionally, we closely monitored the animals’ rectal temperature, which can change rapidly and affect both cardiac and respiratory rates, to ensure that the temperature remained within normal levels throughout the experiments. In vivo data collection used a probe light source–detector separation of 8.0 mm. As a rule of thumb, the measurements were expected to be from a depth equal to around 1/2 to 1/3 of the source–detector distance [30,31]. The experiment with the optical probe showed a 100% difference in the PCA score plot of the data collected from 6 mm and 8 mm source–detector distances [21]. The optical probe previously showed the capability to differentiate post-mortem pork tissues using the source and detector at an 8 mm separation in the PCA score plot [32]. In this experiment, the data collected include ten measurements by positioning the sensor on the animal’s leg above the implant. The optical data are specific to the location from where they were collected in the animal [33]. These biological differences across the body parts call for the model to be specific to the body location.

#### 2.3.3. In Vivo Data Processing

Only data collected from the animals’ right femur were considered for data processing. The first step of processing the data was to separate the required conditions while the spectrometer range with noise was avoided. Different layers, such as fat, skin, muscles, and blood flow, influence the in vivo optical data from animals. The fat scatters light, while hemoglobin in the blood absorbs NIR light [31,34,35]. Due to the degradation of magnesium-based implants, there are complex chemical reactions near the implants, including changes in pH [15,25]. The hydrogen gas formed as part of the implant reaction occupies the regions near it, and these gas bubbles near the implant can slowly increase in dimension as they move close [14]. Such bubbles influence the optical information due to light scattering if they fall in the light’s path. In addition, as the wound heals, there are changes in the skin tissue [30]. The body healing after the surgical procedure adds more changes at the cellular level as damaged cells start to heal in the upcoming days of the post-surgery period. Hence, the optical information gathered from the surface above the implant is the net effect of complex biological changes. Thus, in vivo data need further preprocessing to identify the information regarding the implant surface.

The standard normal variate (SNV), which divides the mean-centered spectrum by its standard deviation, was used to preprocess the in vivo spectra [36,37]. A derivative approach can help to enhance the changes in the spectrum. The Savitsky–Golay filter derivative technique with the first derivative [34] was used after SNV. Following these steps for all datasets, a similar trend was observed in the in vitro data in the PC1–PC2 plot by separating the optical data into different day groups.

## 3. Results

The in vitro experiment aims to study the interaction of implants with DMEM cell culture medium. Figure 3a shows the medium pH values over time. The medium tends to become alkaline over time, primarily due to the implants’ degradation in the medium. The PCA-based score plot from the optical NIR spectra from different days is shown in Figure 3b. The in vitro data are influenced significantly by PC1 as the explained variance is 96.9%, while PC2 is 2.5%. The graph shows that the measurements made on the three implant samples seem to be clustered by day (day 0, day 2, day 5, and day 10). However, on day 5, the sample distribution is drastically influenced. As previously shown by the author in [14], these changes on day 5 are not outliers; there is meaningful information related to the changes in the medium.

The implant interacts with the medium, which leads to an expected change over days. The trend identified is that the samples can be grouped based on the day. The in vitro data analysis resulting in Figure 3a,b suggests that optical data on a particular day change proportionally to the internal changes in the surrounding medium through which the light passes. Here, we explored the hypothesis that the “implant interaction with the medium leads to an expected change over days, measurable with optical spectra”. The same hypothesis was further investigated in the in vivo experiment. The recovery progression due to the magnesium implant (WE43 type) was expected to be similar post-surgery. It has to be noted that the degradation rate of WE43 is different from ZX00 [38]. However, as these two alloys are used in two different settings, the conclusions would not contradict each other as the in vivo pH is controlled locally, and the structural changes would not be the only influencing cause of the detected signals [15]. Slight variations based on the difference in rats is acceptable.

Among the four rats, three rats survived the first two weeks. Rat 1 died on the third day after implant surgery. The optical data for rat 1 differ, whose health was negatively progressing post-surgery. However, the variation in the optical data from rat 2 that survived the first two weeks (Figure 4a,b) challenges the trend of the in vitro experiment. Hence, the reason for the variation in this rat’s optical spectrum needs clarification to confirm the hypothesis of this paper.

A loading plot in the PCA can relate to the features in the datasets [39]. In the spectral datasets, features are the wavelengths. Studying the wavelengths that influence the spectra can be meaningful. The literature illustrates that chromophores influence different wavelengths. The NIR range strongly influences absorption due to hemoglobin and cytochrome C oxidase. Oxyhemoglobin absorption rises, while deoxyhemoglobin reduces below 800 nm, and metabolism-related changes (cytochrome C oxidase) peak at around 800 nm [31,40,41,42]. The region above 940 nm hints at water content in the tissues. The in vivo preprocessing uses the derivative-one filter that helps to highlight the wavelengths that have changes in the raw spectra. The slow rise in the absorption curves is converted as the sharp changing peaks in the derivative curves of spectra. The amplitude of the sharp peaks is proportional to the changes in the actual spectra [40]. The derivative spectra of in vivo data from the rats are studied. The derivative spectra are plotted using measurement 7 (random selection) from a particular rat on a specific day.

Figure 5 is the derivative spectra plotted for all four rats on day 3. It is interesting to note that two rats (rat 3 and rat 4) overlapped. Due to the 100% overlap, only rat 4 is visible in the graph. Referring to the PCA plot from Figure 4 on the third day for rats 3 and 4, it is due to this 100% overlap. It is to be noted that rat 2, which also survived for two weeks, had spectra different from the other surviving rats. It closely resembles the derivative spectra of rat 1 in specific wavelengths.

## 4. Discussion

The findings of the in vivo experiment indicate that utilizing derivative spectra to identify wavelengths contributing to progress in Mg-based implant surgery is an effective approach. Score plots based on PCA can differentiate rats based on their internal changes. The optical data reflect a combined effect of internal changes. The first week after the implant is critical, as illustrated by the pH curve in Figure 3a from the in vitro experiment, and any deviation from the required progression in the optical spectra signals the need for medical attention. The derivative spectra from the in vivo experiment on day 3 (Figure 5) demonstrate that hemoglobin-based changes were opposite for the dead rat (rat 1) and rat 2 with skin rashes compared to the other rats without visible complications. This study confirms the feasibility of applying PCA to optical spectra data to identify deviations in the positive progression of Mg-based implant surgery.

When analyzing the PCA-based score plots in Figure 4a for in vivo data, it is evident that there are two distinct clusters for day 3, separated by 100%. The derivative spectra for the same day in Figure 5 show that rat 3 and rat 4 exhibited similar optical spectra that overlapped. Thus, the optical spectra from rat 3 and rat 4 are one cluster on the positive axis of PC1, while the other cluster is from rat 1 and rat 2 (refer to Figure 4b). Hence, it can be considered a reference to compare the other two rats on day 3. In relation to the derivative spectra results presented in Figure 5, it is observed that rat 1 and rat 2 exhibit an inverse pattern when compared to rat 3 and rat 4 (the rats that survived for two weeks).

**Figure 5 sensors-23-03063-f005:**
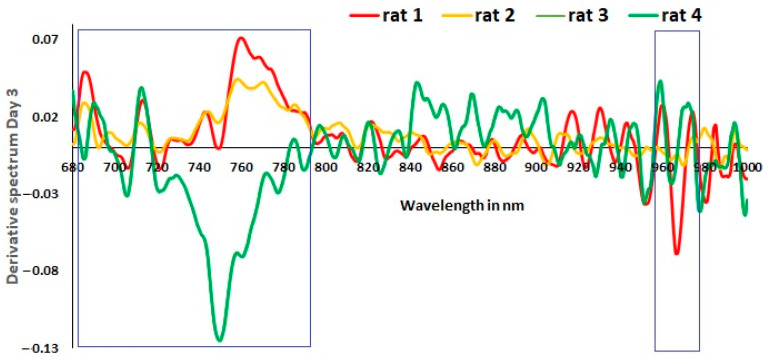
In vivo results. Derivative spectra plotted from one measurement of day 3. It is plotted for all four rats. Rat 3 and rat 4 overlap 100%. Rat 1 and rat 2 are significantly different from rats 3 and 4. Later that same day, rat 1 died. This rat showed two predominant peaks that stand out from the normal rats (rat3 and rat4), one at 965–975 nm and the other at 680–800 nm. The one at 965–975 nm and the other at 680–800 nm correspond to water absorption and hemoglobin.

Interestingly, rat 1, which survived only until day 3, showed two predominant peaks that stand out from the normal rats, one at 965–975 nm and the other at 680–800 nm. The one at 965–975 nm and other at 680–800 nm correspond to water absorption and hemoglobin, respectively [35]. It is evident that rat 2 also has a distinct pattern. For the derivative spectra from days 3 and 14 (Figure 5 and Figure 6), rat 2 shows significant peaks between the wavelengths 680 nm and 800 nm, suggesting issues related to blood flow on the implantation site highlighted in Figure 6. Observational analysis reported red rashes near the wound site, a possible allergic reaction to the shaving cream that led to a continuous inflammatory state of the skin. Hence, we believe this abnormal skin rash influenced the optical measurements and might justify the significant difference near 955 nm compared to rats 3 and 4, as highlighted in Figure 6. Therefore, we hypothesized that the derivative spectra, particularly in the 700–900 nm range collected from rat 1 and rat 2, indicated changes in hemoglobin and cell metabolism at the wound site that correlated with outcomes not associated with the successful progression of healing (reference to Figure 2 in [31]). Optical data depend on local tissue [33]. It is possible that wound healing combined with the effects of magnesium implant reactions in vivo contributed to the optical changes near the interface for rat 2.

A previous in vivo study [14] suggests that by the fifth day, maximum internal changes occur due to wound and implant reactions. It is essential to separate days in the in vivo setup because each time point is associated with a stage of healing. If optical measurements along with PCA can distinguish between days after surgery, it would also mean that they could detect progress in the healing process of the tissue. The two rats with an abnormal response to surgery (rat 1 died, rat 2 developed an allergic skin reaction) presented a very different derivative spectra from the healthy rats (rats 3 and 4). In fact, rats 3 and 4, which presented a normal recovery process after implantation, had overlapping derivative spectra. The spectral analysis for rats 1 and 2 demonstrates the potential of spectral analysis to detect abnormal recovery processes. Although the limited number of samples is a constraint in this pilot study, it highlights the potential for a future NIRS device.

## 5. Conclusions

This research uses exploratory data analysis to relate the optical changes near the in vivo magnesium implant interface. In vitro experiments facilitated the identification of the trend and the possibility of grouping optical data collected from a particular day. They helped to initiate this study from an optically less complicated medium due to the absence of blood flow and cellular metabolism. The in vivo optical data analysis gave meaning to the trend by comparing the health of the rats in the post-surgery days. The possibility of separating days is intriguing in the in vivo studies. The study of two particular cases of rats confirmed that it is possible to separate rats with variations in recovery progression post-surgery optically. The biochemical reactions occurring near the implant site over time led to differences in the optical spectra. The optical probe developed at the university can capture these combined changes in vivo. Hence, optical diagnosis seems to be a promising approach. The dynamic degradation of magnesium implants with simultaneous tissue regeneration requires medical follow-up support for extended periods. The optical probe we have developed offers a noninvasive approach that can be useful for monitoring the early healing stage. It gives information on oxygen saturation, hemoglobin concentration, [28,43] and water content, which may indicate physiological healing processes. Further measurements conducted with the optical probe may lead to a characteristic optical pattern associated with the healing process.

In the future, it can become a user-friendly device and be used by clinicians and patients periodically during post-operative periods where physical movement is restricted. Furthermore, predictive models can be developed based on the knowledge of magnesium implant degradation behavior and the specific optical patterns associated with the process. A deviation from these expected optical patterns may be an indication of an abnormal healing process and an early sign for further medical analysis. On the other hand, patients showing an expected optical pattern from the knowledge base statistics can avoid unnecessary hospitals visits. Additional information on oxygen saturation, hemoglobin, and water hydration can also help relate to a need for a detailed medical check-up.

## Figures and Tables

**Figure 1 sensors-23-03063-f001:**
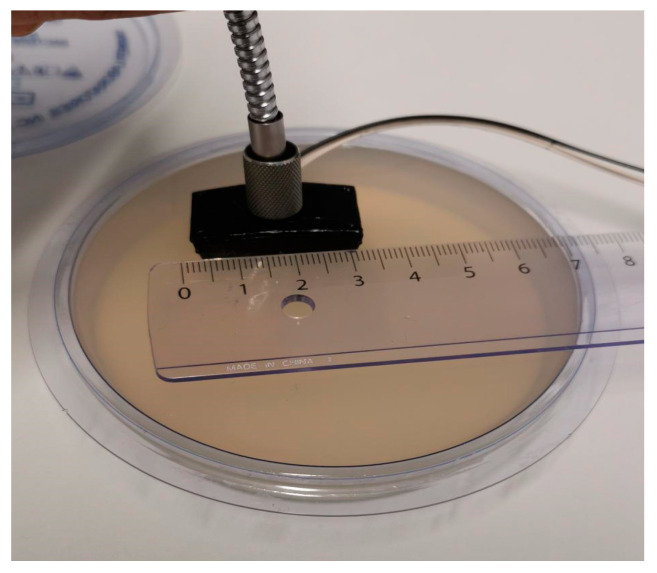
The probe and its optical connection during the in vivo experiment.

**Figure 2 sensors-23-03063-f002:**
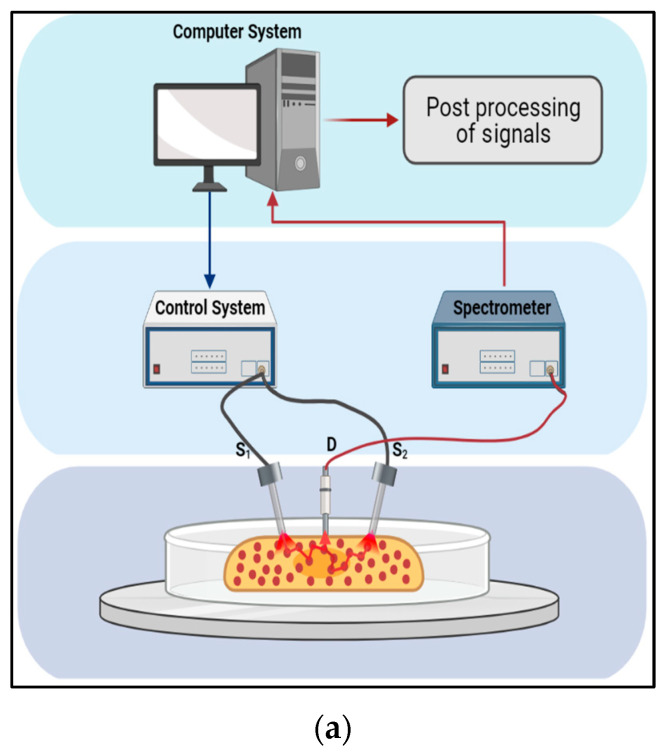
The experiment set up for the optical measurements in this paper: (**a**) Refection-type measurement where the source (S) and detector (D) are on the same side of the medium. It uses the optical probe given in Figure 1. This is the experimental setup used for the in vivo experiment [21]; (**b**) transmission-type measurement where the medium is taken in a cuvette, which is placed in the holder. This is the experimental setup used for in vitro experimentation [25].

**Figure 3 sensors-23-03063-f003:**
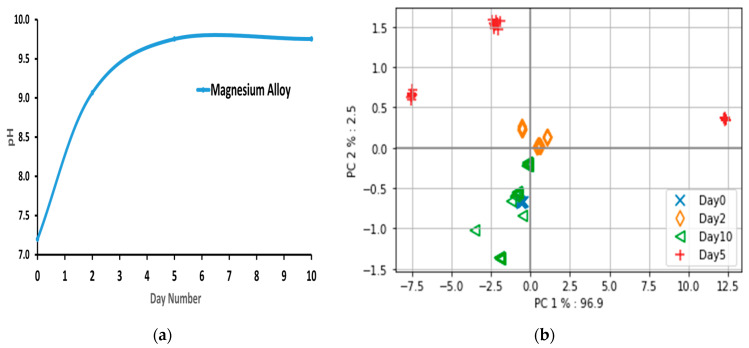
Observations from the in vivo experiment, including (**a**) the medium pH values over time and (**b**) the PCA-based score plot analysis of the in vitro optical data, which indicates that implant interactions with the medium result in a detectable change over days. In contrast to the trend observed in the in vitro experiment, it was not possible to differentiate the days in the in vivo experiment using PCA-based score plots. For instance, in Figure 4a, the day 3 samples exhibit two 100% separate clusters, representing the rats’ optical information on the same day. However, the reason for such different separations requires further investigation. Animals naturally exhibit biological variability that can influence the local physiological response to the implant, leading to differences in the optical spectra and the creation of distinct clusters, unlike in vitro experiments.

**Figure 4 sensors-23-03063-f004:**
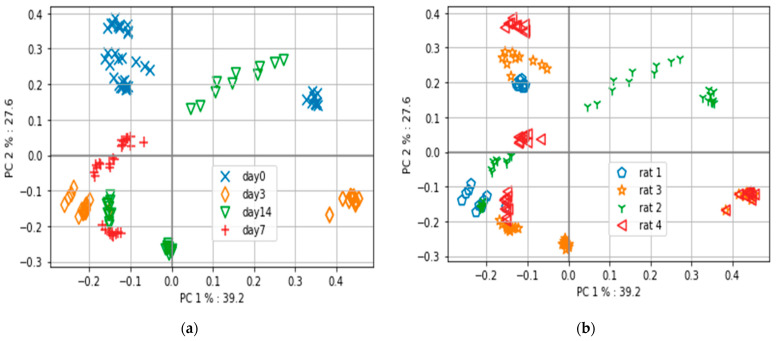
In vivo PCA results. Optical data are collected from rats referred to as 1, 2, 3, and 4 on days 0, 3, 7, and 14: (**a**) PCA plot based on different days; (**b**) PCA plot based on different rats.

**Figure 6 sensors-23-03063-f006:**
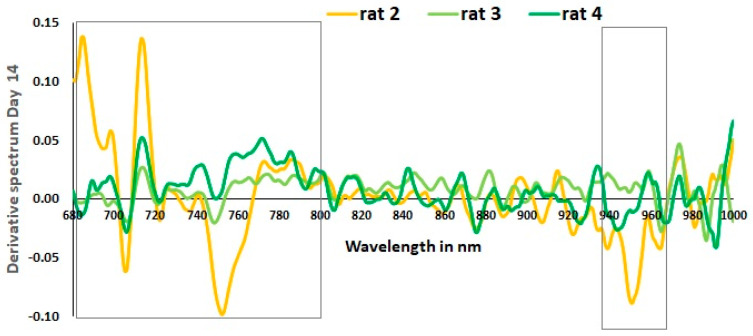
In vivo results. Derivative spectra plotted from one measurement of day 14. It is plotted for all three rats that survived until day 14. Rat 2 has significant peaks compared to the other two rats between the wavelengths 680 nm and 800 nm, suggesting issues related to blood flow on the implantation site highlighted using the first box. The abnormal skin rash in rat 2 influenced the optical measurements near 955 nm compared to rats 3 and 4, highlighted using the second box.

## Data Availability

Raw data are available upon request.

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
