# Peer review of "In Vivo Analysis of a Biodegradable Magnesium Alloy Implant in an Animal Model Using Near-Infrared Spectroscopy"

_sensors, 2023, doi:10.3390/s23063063_

Round 1
Reviewer 1 Report
In this work a specialized optical probe for in vivo studies and in vitro lab experiments was used. Spectroscopic data were acquired for two weeks in vivo and in vitro to study the combined effect of biodegradable Mg-based implant disks on the cell culture medium and in vivo. You do guide readers well through your experiment and results. This is true for the abstract and the introduction. In the materials and methods part you should add some information on your in vivo experiments to give full understanding about your work.
1. How was the positioning of the optical probe done and how did you perform comparable positions in different animals?
2. Did you use anesthesia during the multiple (ten) measurements on each timepoint?
3. (How) did you monitor the physiological state of the animal?
4. Why are there different timepoints in vivo and in vitro, please explain.
I guess you did measure the basic parameters like temperature and breathing rate. Can you please show the stability of these data over time and in between the different animals
The stability and amount of blood flow will influence your NIR measurement. How did you calibrate the sensor for the in vivo experiments?
Did you measure any control rat without implants over time?
The DMEM cell culture can have high or low glucose and added phenolic red color. It looks like phenolic red color in your pictures. Did you notice any color changes when the pH changed during implant degradation? Can you please give the exact DMEM composition.
Principal Component analysis (PCA) is a valuable technique for analyzing large datasets containing a high number of dimensions per observation while increasing the interpretability of data and preserving the maximum amount of information. There are several approaches to perform PCA, like using eigenvectors of the data's covariance matrix. Did you use a validated library in Python or a “homebrew” software?
How did you check your PCA for reliability?
There are a few imprecisions you should brush up:
What do you mean with “data cleaning” in chapter 2.2?
What is "surgery progression" in line 108, page 3?
Probably there are some “copy paste” errors in the text that derange comprehension. Please check page 3, line 108 and line 124
The results show that the optical probe can capture variations in vivo from biological tissues from rats with biodegradable magnesium alloy "WE43" implants. Do you think this might differ with other alloys?
Author Response
We wish to extend our sincere appreciation to the reviewers for their invaluable feedback and thought-provoking inquiries, which have substantially improved the quality of our manuscript. We are meticulously addressing each remark in a methodical and organized approach. please consider our replies in the attached file.

Reviewer 2 Report
Summary: The authors present an exploratory longitudinal study investigating the mange of Biodegradable Magnesium (Mg) based implants to the body tissues in vivo. They used near-infrared (NIR) spectroscopy to assess the magnesium's physiological events. The authors reported the spectral alteration at four-time points after implanting the biodegradable magnesium alloy in the rodent's leg.
Strengths: The clinical question is relevant and well-described. There is an unmet need for non-invasive and sensitive approaches to investigate the physiological interaction of biodegradable implants with tissues. This study demonstrates the potential of NIR spectroscopy for in vivo longitudinal examinations.
Weaknesses: Not clear whether the method could be used in clinical routine; nevertheless, the method's utility for pre-clinical investigations holds promise for further studies.
The study design, implementation, and data analysis are scientifically sound. The conclusion fairly reflects the analyzed data and the provided results. Some minor comments as follow:
Pg 3, Ln 132: Is this the aim of the study? The authors should clarify the study's aims and objectives distinctive from the hypothesis.
Pg 4, Ln 144: Please rephrase this sentence.
Pg 5, Ln 148-156: This format is unusual for several journal formats. This type of structure is more for a thesis report or book chapter. I would suggest removing the subsection addressing, as this is a few pages of an article.
Pg 5, Ln 157: It is unclear why the authors have named this subsection "Methods". The parent section is Materials and Methods. This is not an inclusive title, and I suggest rephrasing it.
Pg 6, Ln 197: Figure 3: invitro -> in vitro.
Pg 6, Ln 199 - 210: this paragraph consists of some results and discussion (as the author points out, the observations are not outliers…). I suggest removing the figure and text to the results and discussion sections.
Discussion: The discussion section is a bit confusing and poorly structured. I suggest starting the discussion section with the text given in the third paragraph, which reflects the main findings more closely than paragraphs 1 and 2. There is some redundancy in paragraphs one and two, speculating that the rashes influenced the optical measurements. Authors may consider rephrasing the two first paragraphs. The limitations of the study are not addressed in the discussion section.
Author Response
We wish to extend our sincere appreciation to the reviewers for their invaluable feedback and thought-provoking inquiries, which have substantially improved the quality of our manuscript. We are meticulously addressing each remark in a methodical and organized approach. Please consider our response in the attached file.

Reviewer 3 Report
Please see the attached file.

Author Response

(The authors gave the same response as above.)

Round 2
Reviewer 3 Report
Recommended for Acceptance